# Multivariate Analysis of Adverse Reactions and Recipient Profiles in COVID-19 Booster Vaccinations: A Prospective Cohort Study

**DOI:** 10.3390/vaccines11101513

**Published:** 2023-09-23

**Authors:** Ryuta Urakawa, Emiko Tanaka Isomura, Kazuhide Matsunaga, Kazumi Kubota

**Affiliations:** 1Department of Pharmacy, Osaka University Dental Hospital, 1-8 Yamada-oka, Suita, Osaka 565-0871, Japan; 2Department of Clinical Pharmacy Research and Education, Graduate School of Pharmaceutical Sciences, Osaka University, 1-6 Yamada-oka, Suita, Osaka 565-0871, Japan; 3Department of Oral and Maxillofacial Surgery, Graduate School of Dentistry, Osaka University, 1-8 Yamada-oka, Suita, Osaka 565-0871, Japan; 4Department of Oral & Maxillofacial Oncology and Surgery, Graduate School of Dentistry, Osaka University, 1-8 Yamada-oka, Suita, Osaka 565-0871, Japan; 5Department of Healthcare Information Management, The University of Tokyo Hospital, 7-3-1 Hongo, Bunkyo-ku, Tokyo 113-8655, Japan

**Keywords:** SARS-CoV-2, age, sex, comorbidities, underlying disease, side effect, vaccination interval, cross vaccination, adverse events following immunization

## Abstract

Background: This study investigated the impact of vaccinated population profiles, vaccine type/interval, and the number of vaccine doses on adverse reactions to receiving a coronavirus disease 2019 (COVID-19) booster vaccination. Methods: A survey of adverse reactions was conducted from January 2022 to December 2022 among Osaka University Dental Hospital employees who received their third or fourth doses. The study included 194 third-dose recipients and 131 fourth-dose recipients. Comparisons of the occurrence of adverse reactions between the third- and fourth-dose groups were analyzed via a chi-squared test. The relationships between each adverse reaction occurrence and recipient profiles, vaccine type/interval, and the number of vaccine doses were analyzed via a logistic regression analysis. Results: No significant differences were found in the occurrence of adverse reactions between the third and fourth doses. Younger recipients often developed injection site reaction, fatigue, chills, fever, arthralgia, headache, diarrhea, and any adverse reactions more often. Females had higher frequencies of fatigue, chills, headache, and nausea compared to males. Recipients without underlying diseases had higher frequencies of fever and nausea than those with underlying diseases. Conclusions: Younger recipients and females were at higher risk for adverse reactions to a COVID-19 booster vaccination, while the number of vaccinations, vaccination interval, vaccine type, and cross-vaccination showed no significant associations.

## 1. Introduction

Viruses generally mutate little by little during repeated multiplications and epidemics. A typical severe acute respiratory syndrome coronavirus 2 (SARS-CoV-2) is estimated to mutate at a rate of about two nucleotides per month in its genome, which is slower than other viruses such as influenza and human immunodeficiency virus (HIV) [1]. Since SARS-CoV-2 has spread worldwide due to the pandemic, it is assumed to infect humans efficiently due to continuous mutations [2]. SARS-CoV-2 can also cause unpredictable and severe outbreaks via the appearance of variants of concern (VOCs), which must be controlled through the development of future vaccines and the improved and strategic use of therapeutic agents [3].

Vaccines, like other medications, carry the potential for adverse events. Therefore, it is crucial that the risks of these adverse events do not outweigh the benefits of vaccination. For example, a report from Italy indicates that while allergic reactions can sometimes lead to anaphylaxis, which is a fatal adverse reaction, the frequency of these incidents is low and is outweighed by the benefits of the vaccine [4]. On the other hand, determining whether an adverse event was caused by a vaccine and establishing causality is of the utmost importance. This can be achieved through epidemiologic studies designed to minimize bias and account for confounding factors. A report on HPV vaccines emphasizes the significance of standardized algorithms, such as the WHO causality assessment, and the use of clinical records [5].

Those diagnosed with coronavirus disease 2019 (COVID-19) who are more susceptible to severe disease include the elderly, immunocompromised individuals, those with underlying diseases, pregnant women, smokers, and obese individuals [6]. Underlying diseases that correlate with higher risks of severe illness from COVID-19 include cancer, chronic kidney disease, chronic liver disease, chronic lung diseases, diabetes, heart conditions, stroke, and cerebrovascular disease [6]. Vaccination has been shown to be effective at reducing severe illness, hospitalization, and death [7].

Various types of vaccines against SARS-CoV-2 are under development in Japan and abroad, and several types are already in clinical use, including mRNA, inactivated virus, viral vector, and protein subunit vaccines [8]. In Japan, the BNT162b2 vaccine from Pfizer was approved, and vaccination began in February 2021; additionally, the mRNA-1273 vaccine from Takeda/Moderna and the AZD1222 vaccine from AstraZeneca vaccine were approved in May 2021. The booster vaccination was initiated in December 2021, using BNT162b2 and mRNA-1273. Those aged 18 and older who had received their second dose at least 8 months earlier were vaccinated with a third dose. Subsequently, the interval between additional vaccinations was shortened to a minimum of 6 months, and the eligible age was expanded to those aged 12 and older. The fourth booster vaccination began in May 2022, and an Omicron-compatible bivalent vaccine was approved in September 2022. Currently, voluntary inoculations with the monovalent Pfizer vaccine COMIRNATY^®^, the bivalent Pfizer vaccine COMIRNATY^®^ Original/Omicron BA.4-5, the bivalent Moderna vaccine SPIKEVAX Original/Omicron BA.4-5, and te NUVAXOVIDTM vaccine are underway, and additional booster doses are available if the recipient was vaccinated at least 3 months after the previous vaccination [9]. Influenza is currently prevalent in Japan along with COVID-19, and the safety and efficacy of simultaneous vaccination with COVID-19 and influenza vaccines has been recognized by the Government of Japan, as have reports on the safety and efficacy of such vaccinations [10], and simultaneous vaccinations are now possible in Japan. Because of the ability of SARS-CoV-2 to evolve rapidly, it is unlikely that it will be eradicated, and we will have to live with this virus for a long time. Although the use of booster vaccinations has proven to be extremely effective against severe infections caused by the Delta and Omicron variants, it is possible that variants with more severe symptoms or more potent immune escape mechanisms may emerge in the future. Thus, the development of next-generation vaccines and continuous booster vaccinations may be necessary [11].

Most adverse reactions to the COVID-19 vaccine reported range from mild to moderate and resolve within a few days. Typical adverse reactions are injection site pain, fever, fatigue, headache, myalgia, chills, and diarrhea [12]. Severe adverse reactions such as anaphylaxis, cardiovascular events, or hematological events have been reported but are very rare [12,13,14]. Despite the proven efficacy of COVID-19 vaccination, there has been significant hesitancy to receive the primary vaccine and booster. The most common reasons for avoiding COVID-19 vaccination include a lack of safety information and a fear of adverse reactions [15]. It is important to be sufficiently aware of the risks of adverse reactions. Studies have shown that adverse reactions to vaccines against SARS-CoV-2 are more likely to occur in young people and females and vary according to the type of vaccine [16]. Our previous studies found similar results, with younger people or females being more likely to develop adverse reactions to the first, second, and third doses of Pfizer vaccines, respectively [17,18]. However, the studies only dealt with cases caused by the monovalent Pfizer vaccine COMIRNATY^®^ and did not consider the effect of vaccine type. The number of vaccination doses and the type/interval of administration should also be included as confounding factors in booster vaccinations that will continue to be promoted. Therefore, based on the hypothesis that the profile of the recipient, the type/interval of vaccine, and number of vaccination doses influence adverse reactions to booster vaccinations with a COVID-19 vaccine, this study conducted a questionnaire survey and analysis of adverse reactions among recipients who received a booster vaccination at Osaka University Dental Hospital.

## 2. Materials and Methods

### 2.1. Study Design

This prospective cohort study included workers at the Osaka University Dental Hospital who received the third and fourth booster doses of a COVID-19 vaccine. Questionnaires on adverse reactions and the vaccinated population profile were conducted from January 2022 to December 2022. Participants were invited to participate in the study on a voluntary basis, and responses to the questionnaire without research consent were excluded. Questionnaires were provided to 617 recipients of the booster vaccine, and 194 recipients of a third dose and 131 recipients of a fourth dose responded to the survey.

### 2.2. Investigation Methods for Adverse Reactions and the Backgrounds of the Recipients

The self-administered survey was conducted anonymously via web forms and questionnaires. The survey items included age, sex, type of vaccine, vaccination interval, underlying disease, and the type and severity of adverse reactions. The type of vaccine consisted of 4 kinds of vaccine: the monovalent Pfizer vaccine (original), the bivalent Pfizer vaccine (BA.1/BA.4-5), the monovalent Moderna vaccine (original), and the bivalent Moderna vaccine (BA.1/BA.4-5), each treated as a distinct vaccine type. Underlying diseases were defined as cardiac disease, renal disease, hematological disease, immunodeficiency, and others. The types of adverse reactions were an injection site reaction, fatigue, chills, fever, arthralgia, myalgia, headache, diarrhea, nausea, and others. The severity of adverse reactions was described based on the Common Terminology Criteria for Adverse Events (CTCAE) Version 5.0 [19]. Each adverse reaction was graded according to the CTCAE Version 5.0. The grade of the worst adverse reaction was defined as the worst grade among all the adverse reactions developed in an individual. The absence of occurrence was defined as Grade 0. Homologous vaccination was defined as the use of the same type of vaccine as the second vaccination for the third vaccination and the same type of vaccine as the third vaccination for the fourth vaccination. Heterologous vaccination was defined as the use of a different type of vaccine for the third or fourth dose compared to the previous dose.

### 2.3. Statistical Analysis

A comparison of the presence or absence of adverse reactions between the third and fourth doses was performed using a chi-squared test. The relationship between the presence/absence of each adverse reaction and the vaccinated population profile was analyzed via a logistic regression analysis. The logistic regression analysis was performed using univariate and multivariate analyses with the number of vaccinations, age by 10 years, gender, the presence of underlying disease, vaccine type, and 30-day intervals between vaccinations used as independent variables and the presence of each adverse reaction taken as the dependent variable. All analyses were performed using SPSS Statistics ver. 26 from IBM, with *p* < 0.05 defined as significantly different on a two-tailed test.

### 2.4. Ethical Review

This study was conducted under the approval of the Ethical Review Committee of Osaka University Dental Hospital (Approval No. R3-E11-3).

## 3. Results

### 3.1. Profile of Study Participants and Adverse Reactions to Vaccinations

The profile of the subjects is shown in Table 1. A total of 325 subjects were included in the study: 194 subjects for the third immunization and 131 subjects for the fourth immunization.

The severity of adverse reactions to the third and fourth doses is shown in Table 2. 

### 3.2. Comparison of Adverse Reactions to the Third and Fourth Doses

A comparison of the presence of adverse reactions to the third and fourth doses is shown in Table 3. There were no significant differences in the adverse reactions that developed after the third and fourth doses.

### 3.3. Association between Adverse Reactions and Recipient Profiles in Booster Vaccination

The results of the univariate and multivariate logistic regression analyses are shown in Table 4. The univariate analysis detected significant differences in age, gender, underlying disease, and many adverse reactions. The multivariate analysis showed that older patients were significantly less likely to have an injection site reaction, fatigue, chills, fever, arthralgia, headache, diarrhea, and any adverse reaction than younger patients (odds ratio (OR): 0.54, 0.74, 0.75, 0.79, 0.74, 0.74, 0.47, and 0.25, respectively). We also found that women were more likely to develop fatigue, chills, headache, and nausea than men (OR: 1.94, 1.84, 3.39, and 2.52, respectively), and those without underlying diseases were more likely to develop fever and nausea than those with such conditions (OR: 0.31 and 0.35). There were no significant susceptible adverse reactions in terms of the number of vaccinations, the type of vaccine administered, or the vaccination interval. All the findings were related simply to the presence of an adverse event, not whether it was severe.

The results of the logistic regression analysis with homologous vaccination or heterologous vaccination (cross-vaccination) as the dependent variable instead of the vaccine type are presented in Table 5. No significant association was detected between homologous vaccination or heterologous vaccination and adverse reactions.

## 4. Discussion

Since the spread of COVID-19 throughout the world, SARS-CoV-2 has continued to mutate and cause repeated waves of infection. Vaccination is one of the most effective measures for preventing infection with COVID-19. Staying up-to-date on COVID-19 vaccines provides protection against SARS-CoV-2 infection and significantly lowers the risk of getting very sick, being hospitalized, or dying from COVID-19 [7].

Overall, studies have shown that adverse reactions to vaccines, including the COVID-19 vaccine, tend to be reported more frequently in females than in males. The reasons for this difference are not entirely clear but may be influenced by hormonal factors, differences in immune response [20,21,22], and genetic factors [21,23]. Differences in immune response with age may also be due to a chronic, low-grade pro-inflammatory state associated with aging and a poor response to vaccination due to aging immunity [22,24,25]. The results of the logistic regression analysis of this study support this idea, with younger individuals or females being significantly more prone to adverse reactions than males. When considered together with the results of previous studies, it is believed that this is consistently the tendency for the COVID-19 vaccine [17,18].

The results of the chi-squared test and logistic regression analysis show that there was no difference in the incidence of adverse reactions between the third and fourth vaccinations, nor was there any difference in the vaccination interval. Therefore, it can be concluded that the number of vaccinations and the interval between booster vaccinations have no relationship to the occurrence of adverse reactions in future booster vaccinations. Furthermore, no difference in vaccine type was detected, and it can be inferred that at least for the Pfizer and Moderna vaccines, there is no significant difference in the occurrence of adverse reactions. However, the types of vaccines used in this study were all mRNA vaccines, and whether the same idea can be applied to different types of vaccines such as inactivated viruses, viral vector, and protein subunit vaccines remains to be further investigated, which we consider a limitation of this study. Because cross-vaccination had no association with adverse reactions, cross-vaccination is recommended without concern. Previous studies have reported that systemic adverse reactions are less frequent in recipients who received homologous vaccinations than heterogeneous vaccinations [26], but the present study suggests that the effect on adverse reactions is less than the effects of age and sex.

This study has several limitations in addition to those related to the vaccine type, as described above. One limitation is the small sample size. Typically, age and vaccination interval should be analyzed as continuous variables, but due to the limited number of cases, we analyzed them as categorical variables. Furthermore, while this study focused on the presence of adverse reactions and patient background, most adverse reactions were mild and transient. It would be interesting to explore whether patient backgrounds have an impact on severe adverse reactions, but the number of cases with serious adverse reactions was too small for analysis. Since this study was conducted at a single center, the sample size was not expected to increase even with an extended study period. Future studies with a larger number of cases are needed to address this issue effectively. Another limitation of this study was the potential for selection bias. Given that this was a single-center study involving healthcare providers, it would be beneficial to conduct multicenter case series studies considering occupational, regional, and racial differences. We hope to see similar studies conducted in various locations and regions in the future.

On a global scale, the report found that women, younger people, less-educated people, and people in countries with more stringent government measures were, on average, more likely to be hesitant to vaccinate [27]. The same report also found that people in countries with high (vs. low) uncertainty avoidance were initially (in late 2020) more likely to be hesitant to vaccinate due to concerns about severe adverse reactions, but this difference decreased as the COVID-19 vaccine became more widely available, suggesting that the cultural aspect of uncertainty avoidance may lead to a hesitancy to vaccinate [27]. We believe that the accumulation of knowledge on adverse reactions, as demonstrated in this study, can provide valuable information about risk and help reduce vaccine hesitancy, particularly in countries with high levels of uncertainty avoidance. Furthermore, it is crucial to divulgate the study results among the general public; this has the potential to dispel misperceptions, reduce vague fears, increase confidence in vaccination, and decrease hesitancy.

Although gender and age affect the occurrence and severity of adverse reactions, it is important to remember that most adverse reactions experienced after COVID-19 vaccination are mild and temporary. In addition, although not yet generally recommended [28,29], there is emerging evidence that the prophylactic administration of paracetamol reduces the incidence and severity of adverse reactions to COVID-19 vaccination [30]. Furthermore, vaccination remains an important means of reducing the risks of serious illness, hospitalization, and death associated with COVID-19, regardless of gender or age. An overseas study reported that the risk of hospitalization for the Omicron variant was one-third that of the Delta variant. This is considered to be the result of the overall lower proportion of people who experienced severe disease as a result of booster vaccinations [31]. However, there is a non-zero possibility that future mutant variants will create a worse scenario in which the disease is more severe and even more infectious and vaccines are less effective. Therefore, in the event of an epidemic involving a mutant variant other than the Omicron variant, it is vital to take the following steps: promptly develop an effective vaccine against the new mutant variant, reduce the number of infections, severe cases, and fatal cases by increasing the vaccination rate, prevent the spread of the epidemic through basic infection control measures during the epidemic, and administer a therapeutic agent for COVID-19 as soon as possible. To achieve this, it is necessary to prepare for potential epidemics by researching vaccines or therapeutic agents, establishing vaccination and medical treatment systems, promptly sharing information with the general public, and disseminating knowledge regarding correct infection control measures.

Conducting studies based on real-world data, such as this study, and making the results public is crucial to reducing unnecessary hesitancy towards vaccines. There is also no doubt that vaccination protects many lives against future emerging mutant strains [32]. COVID-19 will continue to mutate and spread infections in the future, and so, to reduce risk, vaccination should be strategically promoted based on accurate information about vaccines.

## 5. Conclusions

Following a COVID-19 vaccine booster immunization, older recipients were found to have fewer incidences of adverse reactions than younger recipients. Females were found to experience more adverse reaction onsets than males. It was suggested that the number of booster doses, the interval between doses, the vaccine type, and cross-vaccination may not be related to the frequency of adverse reactions in booster doses. Further research on the risks and benefits of vaccination and the publication of the results will hopefully reduce the hesitancy to vaccinate and help prevent the spread of COVID-19.

## Figures and Tables

**Table 1 vaccines-11-01513-t001:** Backgrounds of study participants.

	Third Dose	Fourth Dose
Mean age (SD, range)	45.6 (12.9, 20–75)	47.6 (13.1, 24–78)
Sex		
Male: *n* (%)	76 (39.2)	46 (35.1)
Female: *n* (%)	118 (60.8)	85 (64.9)
Underlying diseases		
Yes: *n* (%)	40 (20.6)	33 (25.4)
No: *n* (%)	154 (79.4)	98 (74.8)
Vaccine type		
Monovalent Pfizer vaccine (original): *n* (%)	178 (91.8)	48 (36.6)
Monovalent Moderna vaccine (original): *n* (%)	16 (8.2)	72 (55.0)
Bivalent Pfizer vaccine (BA.1/BA.4-5): *n* (%)	0	10 (7.6)
Bivalent Moderna vaccine (BA.1/BA.4-5): *n* (%)	0	1 (0.8)
Homologous or heterologous vaccination		
Homologous vaccination: *n* (%)	175 (90.2)	56 (42.7)
Heterologous vaccination: *n* (%)	19 (9.8)	75 (57.3)
Mean interval: days since last vaccination (SD, range)	242.1 (19.3, 183–322)	204.9 (26.4, 151–296)

Underlying diseases were defined as cardiac disease, renal disease, hematological disease, immunodeficiency, and others. SD: standard deviation.

**Table 2 vaccines-11-01513-t002:** Severity of adverse events following the third and fourth doses of COVID-19 vaccines.

	Grade 0	Grade 1	Grade 2	Grade 3
	Third Dose/Fourth Dose *n* (%)	Third Dose/Fourth Dose *n* (%)	Third Dose/Fourth Dose *n* (%)	Third Dose/Fourth Dose *n* (%)
Injection site reaction	23 (11.9)/18 (13.7)	103 (53.1)/67 (51.1)	60 (30.9)/42 (32.1)	8 (4.1)/4 (3.7)
Fatigue	57 (29.4)/41 (31.3)	69 (35.6)/38 (29.0)	51 (26.3)/45 (34.4)	17 (8.8)/7 (5.3)
Chills	137 (70.6)/105 (80.2)	43 (22.2)/24 (18.3)	14 (7.2)/2 (1.5)	
Fever	143 (73.7)/100 (76.3)	41 (21.1)/28 (21.4)	10 (5.2)/3 (2.3)	
Arthralgia	117 (60.3)/83 (63.4)	49 (25.3)/26 (19.8)	21 (10.8)/18 (13.7)	7 (3.6)/4 (3.1)
Myalgia	107 (55.2)/84 (64.1)	61 (31.4)/29 (22.1)	20 (10.3)/18 (13.7)	6 (3.1)/0 (0)
Headache	95 (49.0)/70 (53.4)	57 (29.4)/32 (24.4)	33 (17.0)/23 (17.6)	9 (4.6)/6 (4.6)
Diarrhea	185 (95.4)/126 (96.2)	9 (4.6)/5 (3.8)		
Nausea	160 (82.5)/112 (85.5)	20 (10.3)/11 (8.4)	9 (4.6)/8 (6.1)	5 (2.6)/0 (0)
Worst adverse reaction	10 (5.2)/8 (6.1)	85 (43.8)/55 (42.0)	77 (39.7)/56 (42.7)	22 (11.3)/12 (9.2)

**Table 3 vaccines-11-01513-t003:** Comparison of adverse event occurrences following the third and fourth doses of COVID-19 vaccines.

	Grade 0	Grade 1–3	χ^2^ (df = 1)	OR (95% CI)	*p* Value
	Third Dose/Fourth Dose *n* (%)	Third Dose/Fourth Dose *n* (%)			
Injection site reaction	23 (11.9)/18 (13.7)	171 (88.1)/113 (86.3)	0.25	0.84 (0.44–1.64)	0.62
Fatigue	57 (29.4)/41 (31.3)	137 (70.6)/90 (68.7)	0.14	0.91 (0.56–1.48)	0.71
Chills	137 (70.6)/105 (80.2)	57 (29.4)/26 (19.8)	3.74	0.60 (0.35–1.01)	0.05
Fever	143 (73.7)/100 (76.3)	51 (26.3)/31 (23.7)	0.29	0.87 (0.52–1.45)	0.59
Arthralgia	117 (60.3)/83 (63.4)	77 (39.7)/48 (36.6)	0.31	0.88 (0.56–1.39)	0.58
Myalgia	107 (55.2)/84 (64.1)	87 (44.8)/47 (35.9)	2.60	0.69 (0.44–1.09)	0.11
Headache	95 (49.0)/70 (53.4)	99 (51.0)/61 (46.6)	0.62	0.84 (0.54–1.30)	0.43
Diarrhea	185 (95.4)/126 (96.2)	9 (4.6)/5 (3.8)	0.13	0.82 (0.27–2.49)	0.72
Nausea	160 (82.5)/112 (85.5)	34 (17.5)/19 (14.5)	0.52	0.80 (0.43–1.47)	0.47
Any adverse reaction	10 (5.2)/8 (6.1)	184 (94.8)/123 (93.9)	0.14	0.84 (0.32–2.18)	0.71

Chi-squared tests were performed. OR: odds ratio; CI: confidence interval.

**Table 4 vaccines-11-01513-t004:** Associations between adverse event occurrence following booster vaccination and the backgrounds of the study participants.

	Number of Vaccinations	Age (10+)	Sex	Underlying Diseases	Type of Vaccination	Vaccination Interval (30+)
	OR (95% CI)	OR (95% CI)	OR (95% CI)	OR (95% CI)	OR (95% CI)	OR (95% CI)
Univariate model						
Injection site reaction	0.84 (0.44–1.64)	0.52 ** (0.39–0.68)	2.39 * (1.23–4.64)	0.76 (0.36–1.61)	1.98 (0.94–4.21)	1.29 (0.92–1.81)
Fatigue	0.91 (0.56–1.48)	0.72 ** (0.60–0.87)	2.11 ** (1.30–3.42)	0.72 (0.42–1.26)	0.93 (0.61–1.42)	1.09 (0.85–1.39)
Chills	0.60 (0.35–1.01)	0.69 ** (0.56–0.85)	2.11 ** (1.21–3.69)	0.39 * (0.19–0.80)	0.93 (0.59–1.47)	1.28 (0.98–1.67)
Fever	0.87 (0.52–1.45)	0.72 ** (0.58–0.88)	2.07 * (1.18–3.61)	0.25 ** (0.11–0.57)	1.05 (0.67–1.65)	1.13 (0.87–1.47)
Arthralgia	0.88 (0.56–1.39)	0.70 ** (0.59–0.85)	1.56 (0.98–2.51)	0.44 ** (0.25–0.80)	0.89 (0.59–1.34)	1.12 (0.89–1.42)
Myalgia	0.69 (0.44–1.09)	0.87 (0.73–1.03)	1.49 (0.94–2.37)	0.54 * (0.31–0.94)	0.75 (0.49–1.13)	1.25 (0.99–1.58)
Headache	0.84 (0.54–1.30)	0.67 ** (0.56–0.80)	3.98 ** (2.46–6.46)	0.61 (0.36–1.03)	1.29 (0.87–1.93)	1.34 * (1.06–1.69)
Diarrhea	0.82 (0.27–2.49)	0.51 ** (0.31–0.85)	1.53 (0.47–4.98)	0.26 (0.03–1.99)	1.31 (0.54–3.16)	0.89 (0.51–1.54)
Nausea	0.80 (0.43–1.47)	0.87 (0.69–1.10)	2.63 ** (1.30–5.34)	0.31 * (0.12–0.82)	0.92 (0.53–1.59)	1.22 (0.89–1.66)
Any adverse reaction	0.84 (0.32–2.18)	0.27 ** (0.16–0.45)	4.72 ** (1.64–13.60)	0.34 * (0.13–0.88)	1.64 (0.58–4.66)	1.19 (0.73–1.95)
Multivariate model						
Injection site reaction	0.51 (0.16–1.66)	0.54 ** (0.39–0.73)	1.54 (0.74–3.22)	1.52 (0.65–3.52)	2.05 (0.79–5.35)	0.84 (0.51–1.40)
Fatigue	1.06 (0.46–2.42)	0.74 ** (0.60–0.91)	1.94 * (1.16–3.26)	1.02 (0.56–1.87)	0.73 (0.41–1.30)	0.93 (0.65–1.33)
Chills	0.51 (0.21–1.23)	0.75 * (0.60–0.94)	1.84 * (1.03–3.31)	0.52 (0.25–1.11)	1.14 (0.61–2.11)	0.94 (0.64–1.37)
Fever	0.84 (0.36–2.00)	0.79 * (0.63–0.99)	1.77 (0.99–3.17)	0.31 ** (0.14–0.73)	1.00 (0.54–1.84)	0.94 (0.65–1.38)
Arthralgia	1.09 (0.50–2.36)	0.74 ** (0.61–0.91)	1.35 (0.82–2.23)	0.57 (0.31–1.06)	0.76 (0.44–1.33)	0.98 (0.70–1.38)
Myalgia	1.06 (0.50–2.26)	0.95 (0.78–1.15)	1.45 (0.89–2.37)	0.60 (0.33–1.08)	0.69 (0.40–1.20)	1.17 (0.84–1.63)
Headache	0.89 (0.40–1.97)	0.74 ** (0.60–0.91)	3.39 ** (2.04–5.62)	0.93 (0.51–1.69)	1.14 (0.65–1.98)	1.16 (0.82–1.64)
Diarrhea	0.18 (0.02–1.30)	0.47 ** (0.27–0.82)	1.26 (0.37–4.33)	0.42 (0.05–3.41)	2.35 (0.66–8.41)	0.41 (0.17–1.03)
Nausea	1.12 (0.41–3.03)	1.02 (0.79–1.31)	2.52 * (1.22–5.24)	0.35 * (0.13–0.93)	0.78 (0.38–1.60)	1.18 (0.76–1.81)
Any adverse reaction	0.51 (0.08–3.02)	0.25 ** (0.14–0.46)	2.43 (0.70–8.47)	0.95 (0.31–2.96)	0.86 (0.21–3.54)	0.55 (0.26–1.14)

A logistic regression analysis was performed. OR: odds ratio; CI: confidence interval. The OR and 95% CI of age show the ratio per 10 years of age. The OR and 95% CI of the vaccination interval show the ratio per 30 days. In the univariate model, each adverse event occurrence and each characteristic of the study participants’ backgrounds were analyzed separately, and no adjustments were performed. All the statistics of the multivariate model were adjusted for the number of vaccinations, age, sex, underlying diseases, the type of vaccination, and the vaccination interval. All adverse events were analyzed by categorizing them into presence or absence regardless of their severity. *, *p* < 0.05; **, *p* < 0.01.

**Table 5 vaccines-11-01513-t005:** Associations between adverse event and the backgrounds of study participants with heterologous or homologous Doses (cross-vaccination).

	Number of Vaccinations	Age (10+)	Sex	Underlying Diseases	Cross-Vaccination	Vaccination Interval (30+)
	OR (95% CI)	OR (95% CI)	OR (95% CI)	OR (95% CI)	OR (95% CI)	OR (95% CI)
Multivariate model						
Injection site reaction	0.68 (0.22–2.05)	0.53 ** (0.39–0.72)	1.61 (0.77–3.37)	1.52 (0.66–3.51)	0.64 (0.23–1.75)	0.90 (0.55–1.48)
Fatigue	1.08 (0.51–2.31)	0.73 ** (0.59–0.90)	1.97 * (1.17–3.30)	1.02 (0.56–1.87)	1.64 (0.86–3.15)	0.92 (0.65–1.30)
Chills	0.50 (0.23–1.12)	0.75 * (0.60–0.94)	1.84 * (1.03–3.29)	0.52 (0.25–1.10)	0.81 (0.41–1.60)	0.94 (0.65–1.37)
Fever	0.82 (0.37–1.80)	0.79 * (0.63–0.99)	1.76 (0.98–3.15)	0.31 ** (0.14–0.73)	0.95 (0.48–1.86)	0.94 (0.65–1.36)
Arthralgia	0.91 (0.45–1.85)	0.74 ** (0.61–0.90)	1.31 (0.80–2.17)	0.58 ** (0.31–1.07)	1.10 (0.60–2.02)	0.95 (0.68–1.31)
Myalgia	0.93 (0.46–1.85)	0.94 (0.78–1.14)	1.42 (0.87–2.32)	0.61 * (0.34–1.09)	1.35 (0.74–2.47)	1.13 (0.82–1.55)
Headache	1.04 (0.50–2.15)	0.74 ** (0.60–0.91)	3.47 ** (2.09–5.75)	0.93 (0.51–1.69)	1.07 (0.58–2.00)	1.19 * (0.85–1.67)
Diarrhea	0.28 (0.05–1.55)	0.48 ** (0.28–0.83)	1.29 (0.38–4.37)	0.40 (0.05–3.22)	0.57 (0.15–2.16)	0.46 (0.19–1.11)
Nausea	1.18 (0.48–2.96)	1.01 (0.78–1.31)	2.57 * (1.24–5.34)	0.35 * (0.13–0.93)	1.61 (0.70–3.69)	1.18 (0.77–1.79)
Any adverse reaction	0.90 (0.17–4.84)	0.22 ** (0.12–0.43)	2.85 (0.80–10.20)	0.95 * (0.30–2.97)	3.44 (0.67–17.59)	0.59 (0.29–1.22)

A logistic regression analysis was performed. OR, odds ratio; C, confidence interval. The OR and 95% CI of age show the ratio per 10 years of age. The OR and 95% CI of vaccination interval show the ratio per 30 days. All the statistics of the multivariate model were adjusted for the number of vaccinations, age, sex, underlying diseases, cross-vaccination, and the vaccination interval. *, *p* < 0.05; **, *p* < 0.01.

## Data Availability

The data are contained within the article.

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
