# Peer review of "Multivariate Analysis of Adverse Reactions and Recipient Profiles in COVID-19 Booster Vaccinations: A Prospective Cohort Study"

_vaccines, 2023, doi:10.3390/vaccines11101513_

Round 1

Reviewer 1 Report

Multivariate Analysis of Adverse Reactions and Recipient Pro- 2 files in Booster Vaccination of COVID-19: A Prospective Co- 3 hort Study

The authors, working out of a dental hospital in Osaka Japan, examined the side effect of Covid 19 vaccinations in a cross section of patients. They came up with some interesting conclusions, such as that the number of booster doses, the  interval between doses, vaccine type, and cross-vaccination may not be related to the frequency of adverse reactions in booster doses. 226 The authors should use a structures abstract and indicate key statistics such as the number of individuals recruited into the study.

The methodology was also deficient in these key statistics. We did not know about the total subjects recruited until the result section!

The statistics looks okay. But for the multivariate analysis, please indicate which terms were included, and which was the main association sought.

It is a shame that we have information on ONLY the third and fourth dose.

Were there any deaths with the Vaccination?  

Author Response

Response to Reviewer 1 comments

  1. Summary

We appreciate your time and effort in reviewing our manuscript and providing us with valuable feedback. Below, you will find our detailed responses to your comments and corresponding revisions, which are highlighted in track changes in the resubmitted manuscript.

  1. Questions for General Evaluation and Reviewer’s Evaluation

Does the introduction provide sufficient background and include all relevant references?

Can be improved

Are all the cited references relevant to the research?

Yes

Is the research design appropriate?

Yes

Are the methods adequately described?

Must be improved

Are the results clearly presented?

Yes

Are the conclusions supported by the results?

Yes

Response: Thank you for your evaluation. We revised our manuscript following your comments. We would like you to see the Point-by-point response to Comments and Suggestions for Authors described below.

  1. Point-by-point response to Comments and Suggestions for Authors

Comment:

The authors, working out of a dental hospital in Osaka Japan, examined the side effect of Covid 19 vaccinations in a cross section of patients. They came up with some interesting conclusions, such as that the number of booster doses, the interval between doses, vaccine type, and cross-vaccination may not be related to the frequency of adverse reactions in booster doses. The authors should use a structures abstract and indicate key statistics such as the number of individuals recruited into the study. The methodology was also deficient in these key statistics. We did not know about the total subjects recruited until the result section!

Response:

We revised the abstract to a structured version. Also, we provided the number of study subjects in abstract and the Materials and Methods section.

Line 21: The study included 194 third dose recipients and 131 fourth dose recipients.

Line 119-121: Questionnaires were provided to 617 vaccinated recipients, and 194 recipients of third dose and 131 recipients of fourth dose responded to the survey.)

Comment:

The statistics looks okay. But for the multivariate analysis, please indicate which terms were included, and which was the main association sought.

Response:

I apologize for any confusion. I’m afraid, but the information on which terms were included is given in Lines 144-148 of Materials and Methods section: “Logistic regression analysis was performed using univariate and multivariate analysis with the number of vaccinations, age by 10 years, gender, presence of underlying disease, vaccine type, and 30-day intervals between vaccinations as independent variables and the presence of each adverse reaction as the dependent variable.”

The main associations identified are described in Lines 182-191 of Results section: “Univariate analysis detected significant differences in age, gender, underlying disease, and many adverse reactions. Multivariate analysis showed that older patients were significantly less likely to have injection site reaction, fatigue, chills, fever, arthralgia, headache, diarrhea, and Any adverse reaction than younger patients (odds ratio (OR): 0.54, 0.74, 0.75, 0.79, 0.74, 0.74, 0.47, and 0.25, respectively). We also found that women were more likely to develop fatigue, chills, headache, and nausea than men (OR: 1.94, 1.84, 3.39, and 2.52, respectively), and those without underlying diseases were more likely to develop fever and nausea than those with such conditions (OR: 0.31 and 0.35). There were no significant susceptible adverse reactions in terms of number of vaccinations, type of vaccine administered, or vaccination interval.”

If the reviewer mentioned additions to the Abstract section, we have included them in Lines 26-29: “Younger recipients more often developed injection site reaction, fatigue, chills, fever, arthralgia, headache, diarrhea, and Any adverse reactions. Females had higher frequencies of fatigue, chills, headache, and nausea compared to males. Recipients without underlying diseases had higher frequencies of fever and nausea than those with underlying diseases.” We would like to omit further description due to character limitation.

Comment:

It is a shame that we have information on ONLY the third and fourth dose.

Response:

We understand that information on the fifth and higher doses would provide a more useful understanding of booster vaccination, but we finished this questionnaire survey because staff responses have decreased over time. As for the first and second dose which are not booster doses, we previously reported.

Comment:

Were there any deaths with the Vaccination?

Response:

No, there were no vaccination-related deaths.

  1. Response to Comments on the Quality of English Language

Reviewer’s Evaluation: English language fine. No issues detected

Response: Thank you for your evaluation. In accordance with another reviewer’s comment, we consult again a paid editing service, and acquired a minor revision.

Reviewer 2 Report

The authors have reported a fine work. Only minor modifications are necessary before publication.

If possible, provide a structured abstract.

The conclusion section should be carefully modified

There are a lot of statistical values in this manuscript. I advise the authors to recheck all the values.

Try to provide your views about improving the scenario and what measures can be taken to prevent another pandemic.

Some minor grammatical errors are there which needs to be rectified.

Minor editing of the English language is required.

Author Response

Response to Reviewer 2 comments

  1. Summary

We appreciate your time and effort in reviewing our manuscript and providing us with valuable feedback. Below, you will find our detailed responses to your comments and corresponding revisions, which are highlighted in track changes in the resubmitted manuscript.

  1. Questions for General Evaluation and Reviewer’s Evaluation

Does the introduction provide sufficient background and include all relevant references?

Can be improved

Are all the cited references relevant to the research?

Can be improved

Is the research design appropriate?

Can be improved

Are the methods adequately described?

Can be improved

Are the results clearly presented?

Can be improved

Are the conclusions supported by the results?

Can be improved

Response: Thank you for your evaluation. We revised our manuscript following your comments. We would like you to see the Point-by-point response to Comments and Suggestions for Authors described below.

  1. Point-by-point response to Comments and Suggestions for Authors

Comment:

If possible, provide a structured abstract.

Response:

We revised the abstract to a structured version.

Comment:

The conclusion section should be carefully modified

Response:

We reviewed the conclusion section again and revised it.

Original manuscript: In COVID-19 vaccine booster immunization, older recipients were found to have less incidence of adverse reactions than younger recipients. Females were found to have more adverse reaction onset than males. It was suggested that the number of booster doses, the interval between doses, vaccine type, and cross-vaccination may not be related to the frequency of adverse reactions in booster doses.

Revised manuscript: In COVID-19 vaccine booster immunization, older recipients were found to have less incidence of adverse reactions than younger recipients. Females were found to have more adverse reaction onset than males. It was suggested that the number of booster doses, the interval between doses, vaccine type, and cross-vaccination may not be related to the frequency of adverse reactions in booster doses. Further research on the risks and benefits of vaccination and the publication of the results will hopefully reduce hesitancy to vaccinate and help prevent the spread of COVID-19.

Comment:

There are a lot of statistical values in this manuscript. I advise the authors to recheck all the values.

Response:

We thank the reviewer for kind advice. We rechecked all the values and confirmed that there was no error.

Comment:

Try to provide your views about improving the scenario and what measures can be taken to prevent another pandemic.

Response:

According to your suggestion, we added our views about improving the scenario and what measures can be taken to prevent another pandemic in the Discussion section. (Line 282-295: An overseas study reported that the risk of hospitalization for the Omicron variant was one-third that of the Delta variant. This is considered to be the result of the overall lower proportion of people who become severe disease as a result of booster vaccination [27]. However, there is a non-zero possibility that future mutant variants will have a worse scenario: more severe, even more infectious, and less effective vaccines. Therefore, in the event of an epidemic involving a mutant variant other than the Omicron variant, it is vital to take the following steps: promptly develop an effective vaccine against the new mutant variant, reduce the number of infections, severe cases, and fatal cases by increasing the vaccination rate, prevent the spread of the epidemic through basic infection control measures during the epidemic, and administer a therapeutic agent for COVID-19 as soon as possible. To achieve this, it is necessary to prepare for potential epidemics by researching vaccines or therapeutic agents, establishing vaccination and medical treatment systems, promptly sharing information with the general public, and disseminating knowledge regarding correct infection control measures.)

Comment:

Some minor grammatical errors are there which needs to be rectified.

Response:

We consult again a paid editing service and acquired a minor revision. We would like the reviewer to tell us if there are any other errors.

  1. Response to Comments on the Quality of English Language

Reviewer’s Evaluation: Minor editing of the English language is required.

Response: Thank you for your evaluation. We consult again a paid editing service and acquired a minor revision. We would like you to tell us if there are any other errors.

Reviewer 3 Report

The authors did a survey of adverse reaction following the COVID-19 vaccination from January 2022 to December 19 2022 among Osaka University Dental Hospital employees who received the third or fourth doses. The manuscript is poorly written and the results are not well explained. In addition, the area is limited, should include more places/provnicess and other indicuual clusters along with the hospital employess. In addition, a year is not an indicator to conclude the adverse effect following  vaccination, therefore too many questions have to be addressed to be answered at the end of this study.

English is very poor especially the grammar side

Author Response

Response to Reviewer 3 comments

  1. Summary

We appreciate your time and effort in reviewing our manuscript and providing us with valuable feedback. Below, you will find our detailed responses to your comments and corresponding revisions, which are highlighted in track changes in the resubmitted manuscript.

  1. Questions for General Evaluation and Reviewer’s Evaluation

Does the introduction provide sufficient background and include all relevant references?

Must be improved

Are all the cited references relevant to the research?

Must be improved

Is the research design appropriate?

Must be improved

Are the methods adequately described?

Must be improved

Are the results clearly presented?

Must be improved

Are the conclusions supported by the results?

Must be improved

Response: Thank you for your evaluation. We revised our manuscript following your comments. We would like you to see the Point-by-point response to Comments and Suggestions for Authors described below.

  1. Point-by-point response to Comments and Suggestions for Authors

Comment

The authors did a survey of adverse reaction following the COVID-19 vaccination from January 2022 to December 19 2022 among Osaka University Dental Hospital employees who received the third or fourth doses. The manuscript is poorly written and the results are not well explained.

Response:

According to the reviewer’s point, we consult again a paid editing service, and acquired a minor revision. We would like the reviewer to tell us if there are any other errors.

Comment:

In addition, the area is limited, should include more places/provnicess and other indicuual clusters along with the hospital employess.

In addition, a year is not an indicator to conclude the adverse effect following vaccination, therefore too many questions have to be addressed to be answered at the end of this study.

Response:

We revise the discussion section to add some limitations regarded as the reviewer’s comment. (Line 247-260)

As for the area, we added the following description. (Line 256-260)

“Another limitation of this study is the potential for selection bias. Given that this is a single-center study involving healthcare providers, it would be beneficial to conduct multicenter case series studies, considering occupational, regional, and racial differences. We hope to see similar studies conducted in various locations and regions in the future.”

As for the period, we added the following description. (Line 248-256)

“One limitation is the small sample size. Typically, age and vaccination interval should be analyzed as continuous variables, but due to the limited number of cases, we analyzed them as categorical variables. Furthermore, while this study focused on the presence of adverse reactions and patient background, most adverse reactions were mild and transient. It would be interesting to explore whether patient backgrounds have an impact on severe adverse reactions, but the number of cases with serious adverse reactions was too small for analysis. Since this study was conducted at a single center, the sample size was not expected to increase even with an extended study period. Future studies with a larger number of cases are needed to address this issue effectively.”

  1. Response to Comments on the Quality of English Language

Reviewer’s Evaluation: Extensive editing of English language required

Response: Thank you for your evaluation. We consult again a paid editing service and acquired a minor revision. We would like you to tell us if there are any other errors with specific examples.

Reviewer 4 Report

Reasonably good study design but too much reliance on statistical significance and not on the data themselves.  Statistical analysis are not particularly meaningful 

·         P-values shown in table 3 are not informative.   It would be much more informative to show confidence intervals around differences observed to illustrate the variability in the estimates.

·    Table 4 describes “univariate” logistic regression models and multivariate models.  Perhaps it is just the English but not clear what these mean.  More important, 

o   There is no reason that continuous variables like age and vaccination interval should be categorized (in 10 year age groups and increments of 30 days for vaccination interval) since this categorization loses information.  The use of this logistic regression forces a distribution on these variables that may not exist, and isn’t necessary.

o   Also it appears that the outcome variable combines all grades of ADR, which is not very informative since many of these complains are minor and transient.  Readers will be more concerned about persistent and serious ADR.  The data are here - they just weren't used.

o   "Type of vaccination" needs explaining since this study includes 2 manufacturers, 2 types of vaccine (monovalent and bivalent) as well as mixed vaccine receipt.

o   It was interesting that several ADRs appeared on longer intervals but vaccine interval was controlled in multivariate model so hard to interpret

o   Heterologous vs homologous vaccination is hard to evaluate since it is not clear whether this refers to differences between 2nd and 3rd dose or 3rd and 4th

·         It would be desirable to have separate descriptions of the ADR among the 11 people who received bivalent booster should be described separately and not mixed with those receiving a monovalent booster.

pretty good but the explanation of the statistical models is not clear.

Author Response

Response to Reviewer 4 comments

  1. Summary

We appreciate your time and effort in reviewing our manuscript and providing us with valuable feedback. Below, you will find our detailed responses to your comments and corresponding revisions, which are highlighted in track changes in the resubmitted manuscript.

  1. Questions for General Evaluation and Reviewer’s Evaluation

Does the introduction provide sufficient background and include all relevant references?

Yes

Are all the cited references relevant to the research?

Yes

Is the research design appropriate?

Yes

Are the methods adequately described?

Must be improved

Are the results clearly presented?

Must be improved

Are the conclusions supported by the results?

Must be improved

Response: Thank you for your evaluation. We revised our manuscript following your comments. We would like you to see the Point-by-point response to Comments and Suggestions for Authors described below.

  1. Point-by-point response to Comments and Suggestions for Authors

Comment:

Reasonably good study design but too much reliance on statistical significance and not on the data themselves. Statistical analysis are not particularly meaningful.

Response:

The statistical analyses and their interpretation were discussed again with the co-authors' statistical experts, and the manuscript has been revised in accordance with your comments. We would be grateful if you could review them.

Comment:

  • P-values shown in table 3 are not informative. It would be much more informative to show confidence intervals around differences observed to illustrate the variability in the estimates.

Response:

We added 95% confidence intervals to table 3 in response to the reviewer's comments.

Comment:

  • Table 4 describes “univariate” logistic regression models and multivariate models. Perhaps it is just the English but not clear what these mean.

Response:

We revised the footnotes of the Table 4 as follows.

Original manuscript: All the statistics of multivariate model were adjusted for number of vaccinations, age, sex, underlying diseases, type of vaccination, and vaccination interval.

Revised manuscript: In the univariate model, each adverse event occurrence and each characteristic of study participants’ backgrounds were analyzed separately, and no adjustments were performed. All the statistics of multivariate model were adjusted for number of vaccinations, age, sex, underlying diseases, type of vaccination, and vaccination interval.

Comment:

o   There is no reason that continuous variables like age and vaccination interval should be categorized (in 10 year age groups and increments of 30 days for vaccination interval) since this categorization loses information.  The use of this logistic regression forces a distribution on these variables that may not exist, and isn’t necessary.

Response:

Because of the limited number of cases, categorical variables were used in this study. In future studies, when the number of cases increases, it will be necessary to perform statistics using continuous variables, and this has been noted in the Limitation. (Line248-256: One limitation is the small sample size. Typically, age and vaccination interval should be analyzed as continuous variables, but due to the limited number of cases, we analyzed them as categorical variables. Furthermore, while this study focused on the presence of adverse reactions and patient background, most adverse reactions were mild and transient. It would be interesting to explore whether patient backgrounds have an impact on severe adverse reactions, but the number of cases with serious adverse reactions was too small for analysis. Since this study was conducted at a single center, the sample size was not expected to increase even with an extended study period. Future studies with a larger number of cases are needed to address this issue effectively.)

Comment:

o   Also it appears that the outcome variable combines all grades of ADR, which is not very informative since many of these complains are minor and transient. Readers will be more concerned about persistent and serious ADR. The data are here - they just weren't used.

Response:

I understand the reviewer's point, but the number of samples that developed serious ADR was too small. We added a description of the limitations of the sample size. (Line248-256)

Comment:

o   "Type of vaccination" needs explaining since this study includes 2 manufacturers, 2 types of vaccine (monovalent and bivalent) as well as mixed vaccine receipt.

Response:

We thank the reviewer for pointing this out. We added explanation of type of vaccine to the Materials and Methods section. (Line 125-128: Type of vaccine consisted of 4 kinds of vaccine; Pfizer monovalent vaccine (original), Pfizer bivalent vaccine (BA.1/BA.4-5), Moderna monovalent vaccine (original), and Moderna bivalent vaccine (BA.1/BA.4-5), each treated as distinct vaccine types.)

Comment:

o   It was interesting that several ADRs appeared on longer intervals but vaccine interval was controlled in multivariate model so hard to interpret

Response:

The odds ratio of vaccination intervals showed how many times more adverse reactions develop when the 30-day vaccination interval is increased. Because of the limited number of cases, categorical variables were used in this study. In future studies, when the number of cases increases, it will be necessary to perform statistics using continuous variables, and this has been noted in the Limitation. (Line248-256)

Comment:

o   Heterologous vs homologous vaccination is hard to evaluate since it is not clear whether this refers to differences between 2nd and 3rd dose or 3rd and 4th

Response:

We added the definition statement of “heterologous” and “homologous” vaccination in the Materials and Methods and rephrased the word “allogeneic” to “homologous”. (Line 135-139: Homologous vaccination" was defined as the use of the same vaccine as the second vaccination in the third vaccination and the same vaccine as the third vaccination in the fourth vaccination. Heterologous vaccination" was defined as the use of a different vaccine in the third or fourth dose compared to the previous dose.)

Comment:

  • It would be desirable to have separate descriptions of the ADR among the 11 people who received bivalent booster should be described separately and not mixed with those receiving a monovalent booster.

Response:

As shown in Table 1, we treated 11 people who received bivalent booster separately and analyzed as different type of vaccination in Table 4. Also, bivalent vaccines are treated as different type from monovalent in Table 5. If you point out a different intention, please let us know.

  1. Response to Comments on the Quality of English Language

Reviewer’s Evaluation: pretty good but the explanation of the statistical models is not clear.

Response: Thank you for your evaluation. We consult again a paid editing service and acquired a minor revision. Also, we revised the explanation of the statistical modes as described above. We would like you to tell us if there are any other errors.

Reviewer 5 Report

After reading it, I have the following comments:

1.     The article is a sound, well-written example of post-marketing vaccine surveillance. It also comes with a fortunate timing, since COVID-19 booster vaccination campaign will be starting soon in most countries.

2.     Among the article’s keywords, “sex” could be substituted with “AEFIs” or “adverse events following immunization”, which would help readers find it in medical databases.

3.     Authors should include a paragraph to clarify risk of adverse events after immiunization and importance to verify causality assessment

·      doi: 10.3389/fimmu.2023.1074246

·      doi: 10.1016/j.vaccine.2017.03.035

·      doi: 10.1016/j.vaccine.2018.01.018

4.     Line 38: “letters” should be changed into “nucleotides”.

5.     Lines 41-43: “such as” is not use correctly; “via the appearance of” would be better.

6.     Lines 45-47 and 48-50 require references.

7.     The authors should provide context about the COVID-19 booster vaccination offer in Japan. It is also important to include a short description of this vaccine’s schedule in the country: a focus on medium-long term of vaccine effectiveness should be included

·      doi: 10.1016/j.vaccine.2023.07.043

·      doi: 10.1186/s12929-022-00853-8.

8.     The study appears not to be prospective, as data was gathered retrospectively via a self-administered survey and enrolment was done on a different instance than vaccination itself. The authors should clarify, and change nomenclature if needed.

9.     Lines 96-98: the CTCAE’s authors should be referenced.

10.  Lines 125-127: these sentences should be moved to the Materials and Methods section.

11.  “Heterologous” and “homologous” vaccination should be defined in the Materials and Methods section. I also suggest using this terminology uniformly throughout the article (in line 200, “allogeneic” is used).

12.  The authors mentioned vaccine hesitancy, which is indeed a very important topic in present-day vaccinology. However, this topic was neither further described in the Discussion section, nor in the Conclusions section. I suggest to add some considerations about the importance of divulgation of surveillance’s results among the general public, which has the potential to increase confidence in vaccination and decrease hesitancy.

Author Response

Response to Reviewer 5 comments

  1. Summary

We appreciate your time and effort in reviewing our manuscript and providing us with valuable feedback. Below, you will find our detailed responses to your comments and corresponding revisions, which are highlighted in track changes in the resubmitted manuscript.

  1. Questions for General Evaluation and Reviewer’s Evaluation

Does the introduction provide sufficient background and include all relevant references?

Can be improved

Are all the cited references relevant to the research?

Yes

Is the research design appropriate?

Yes

Are the methods adequately described?

Yes

Are the results clearly presented?

Can be improved

Are the conclusions supported by the results?

Can be improved

Response: Thank you for your evaluation. We revised our manuscript following your comments. We would like you to see the Point-by-point response to Comments and Suggestions for Authors described below.

  1. Point-by-point response to Comments and Suggestions for Authors

Comment:

  1. The article is a sound, well-written example of post-marketing vaccine surveillance. It also comes with a fortunate timing, since COVID-19 booster vaccination campaign will be starting soon in most countries.

Response:

We thank the reviewer for warm and encouraging words.

Comment:

  1. Among the article’s keywords, “sex” could be substituted with “AEFIs” or “adverse events following immunization”, which would help readers find it in medical databases.

Response:

Following the reviewer’s suggestion, we add “adverse events following immunization” to the article’s keywords.

Comment:

  1. Authors should include a paragraph to clarify risk of adverse events after immunization and importance to verify causality assessment

  • doi: 10.3389/fimmu.2023.1074246

  • doi: 10.1016/j.vaccine.2017.03.035

  • doi: 10.1016/j.vaccine.2018.01.018

Response:

According to the reviewer’s comment, we add a paragraph to clarify risk of adverse events after immunization and importance to verify causality assessment. (Line 47-56: Vaccines, like other medications, carry the potential for adverse events. Therefore, it is crucial that the risks of these adverse events do not outweigh the benefits of vaccination. For example, a report from Italy indicates that while allergic reactions can sometimes lead to anaphylaxis which is a fatal adverse reaction, their frequency is low and is outweighed by the benefits of the vaccine [4]. On the other hand, determining whether an adverse event was caused by a vaccine and establishing causality is of utmost importance. This can be achieved through epidemiologic studies designed to minimize bias and account for confounding factors. A report on HPV vaccines emphasizes the significance of standardized algorithms, such as the WHO causality assessment, and the use of clinical records [5].)

Comment:

  1. Line 38: “letters” should be changed into “nucleotides”.
  2. Lines 41-43: “such as” is not use correctly; “via the appearance of” would be better.

Response:

We revised as the reviewer pointed out.

Comment:

  1. Lines 45-47 and 48-50 require references.

Response:

We added the following references.

Centers for Disease Control and Prevention. People with Certain Medical Conditions. https://www.cdc.gov/coronavirus/2019-ncov/need-extra-precautions/people-with-medical-conditions.html. Accessed September 8, 2023.

Comment:

  1. The authors should provide context about the COVID-19 booster vaccination offer in Japan. It is also important to include a short description of this vaccine’s schedule in the country: a focus on medium-long term of vaccine effectiveness should be included

  • doi: 10.1016/j.vaccine.2023.07.043

  • doi: 10.1186/s12929-022-00853-8.

Response:

We added the COVID-19 booster vaccination offer in Japan and vaccine’s schedule including a focus on medium-long term of vaccine effectiveness. (Line 66-86)

Revised manuscript: Various types of vaccines against SARS-CoV-2 are under development in Japan and abroad, and several types are already in clinical use, including mRNA, inactivated virus, viral vector, and protein subunit [8]. In Japan, BNT162b2 from Pfizer was approved and vaccination began in February 2021, and mRNA-1273 from Takeda/Moderna and AZD1222 from AstraZeneca vaccine were approved in May 2021. The booster vaccination was initiated in December 2021 using BNT162b2 and mRNA-1273. Those aged 18 and older, who had received their second dose at least 8 months earlier, were vaccinated with third dose. Subsequently, the interval between additional vaccinations was shortened to a minimum of 6 months, and the eligible age was expanded to those aged 12 and older. The fourth booster vaccination began in May 2022, and Omicron-compatible bivalent vaccine was approved in September 2022. Currently, voluntary inoculation with Pfizer monovalent vaccine COMIRNATY®, Pfizer bivalent vaccine COMIRNATY® Original/Omicron BA.4-5, Moderna bivalent vaccine SPIKEVAX Original/Omicron BA.4-5, and NUVAXOVIDTM is underway, and additional booster doses are available if the recipient has been vaccinated at least 3 months after the previous vaccination [9]. Because of the SARS-Cov-2 virus's ability to evolve rapidly, it is unlikely that it will be eradicated, and we will have to live with this virus for a long time. Although booster vaccination has proven to be extremely effective against severe infections caused by Delta and Omicron variants, it is possible that variants with more severe symptoms or more potent immune escape mechanisms may emerge in the future. Thus, the development of next-generation vaccines and continuous booster vaccination may be necessary [10].

Comment:

  1. The study appears not to be prospective, as data was gathered retrospectively via a self-administered survey and enrolment was done on a different instance than vaccination itself. The authors should clarify, and change nomenclature if needed.

Response:

Although this study is a self-administered survey, the questionnaire was distributed at (or before) the time of vaccination, and vaccine recipients responded with a statement of any adverse reactions that had occurred to them. Thus, we consider this study a prospective study.

Comment:

  1. Lines 96-98: the CTCAE’s authors should be referenced.

Response:

We added the following reference of CTCAE.

National Cancer Institute. Common Terminology Criteria for Adverse Events (CTCAE) Version 5.0. https://ctep.cancer.gov/protocoldevelopment/electronic_applications/docs/CTCAE_v5_Quick_Reference_8.5x11.pdf. Accessed September 10, 2023.

Comment:

  1. Lines 125-127: these sentences should be moved to the Materials and Methods section.

Response:

We moved the sentences to the Materials and Methods section in accordance with the reviewer’s comment. (Line 133-135)

Comment:

  1. “Heterologous” and “homologous” vaccination should be defined in the Materials and Methods section. I also suggest using this terminology uniformly throughout the article (in line 200, “allogeneic” is used).

Response:

We added the definition statement of “heterologous” and “homologous” vaccination in the Materials and Methods and rephrased the word “allogeneic” to “homologous”. (Line 135-139: Homologous vaccination" was defined as the use of the same vaccine as the second vaccination in the third vaccination and the same vaccine as the third vaccination in the fourth vaccination. Heterologous vaccination" was defined as the use of a different vaccine in the third or fourth dose compared to the previous dose.)

Comment:

  1. The authors mentioned vaccine hesitancy, which is indeed a very important topic in present-day vaccinology. However, this topic was neither further described in the Discussion section, nor in the Conclusions section. I suggest to add some considerations about the importance of divulgation of surveillance’s results among the general public, which has the potential to increase confidence in vaccination and decrease hesitancy.

Response:

As the reviewer pointed out, we recognize that vaccine hesitancy is a very important topic. We revised the considerations about vaccine hesitancy in the Discussion section and Conclusions section to state the importance of divulgation of surveillance’s results among the general public.

(Line 271-276: We believe that the accumulation of knowledge on adverse reactions, as demonstrated in this study, can provide valuable information about risk and help reduce vaccine hesitancy, particularly in countries with high uncertainty avoidance. Furthermore, it is crucial to divulgate the study results among the general public, which has the potential to dispel misperceptions, reduce vague fears, increase confidence in vaccination, and decrease hesitancy.

Line 304-306: Further research on the risks and benefits of vaccination and the publication of the results will hopefully reduce hesitancy to vaccinate and help prevent the spread of COVID-19.)

  1. Response to Comments on the Quality of English Language

Reviewer’s Evaluation: English language fine. No issues detected

Response: Thank you for your evaluation. We consult again a paid editing service and acquired a minor revision.

Round 2

Reviewer 4 Report

I appreciate the analytic updates. It is much more informative with the Odds Ratios and 95% confidence intervals

For table 4, please add to the footnotes that these data combine all adverse events regardless of severity.  I would have liked to see a sensitivity analysis restricted to severe AEs but realize you may not have enough for such an analysis.   It's important to emphasize in the results that these interpretations (e.g., age effects) relate simply to the presence of an adverse event, not whether it was severe.
Also please consider removing univariate analysis.  It's hard to see what this adds since the multivariate data is much more informative.  also, I found the p-value footnotes irrelevant since the Confidence intervals are so much more informative.  

Author Response

Response to Reviewer 4 comments

  1. Summary

We appreciate your time and effort in reviewing our manuscript and providing us with valuable feedback. Below, you will find our detailed responses to your comments and corresponding revisions, which are highlighted in track changes in the resubmitted manuscript.

  1. Questions for General Evaluation and Reviewer’s Evaluation

Does the introduction provide sufficient background and include all relevant references?

Yes

Are all the cited references relevant to the research?

Yes

Is the research design appropriate?

Yes

Are the methods adequately described?

Yes

Are the results clearly presented?

Yes

Are the conclusions supported by the results?

Yes

Response: Thank you again for your evaluation. We revised our manuscript following your comments. We would like you to see the Point-by-point response to Comments and Suggestions for Authors described below.

  1. Point-by-point response to Comments and Suggestions for Authors

Comment:

I appreciate the analytic updates. It is much more informative with the Odds Ratios and 95% confidence intervals

Response:

We appreciate you for your valuable feedback.

Comment:

For table 4, please add to the footnotes that these data combine all adverse events regardless of severity.  I would have liked to see a sensitivity analysis restricted to severe AEs but realize you may not have enough for such an analysis. It's important to emphasize in the results that these interpretations (e.g., age effects) relate simply to the presence of an adverse event, not whether it was severe.

Response:

We added the following description in the footnotes of Table 4.

“All adverse events were analyzed by categorizing them into presence or absence regardless of their severity.”

Also, we added the following description in the Results section to emphasize that all findings relate simply to the presence of an adverse event, not whether it was severe.

“All the findings were related simply to the presence of an adverse event, not whether it was severe.”

Comment:

Also please consider removing univariate analysis.  It's hard to see what this adds since the multivariate data is much more informative.

Response:

We sincerely appreciate you for pointing this out. We noticed a very important mis-statement. We have amended the labels in Table 5 as follows.

Original manuscript: “Univariate model”

Revised manuscript: “Multivariate model”

As for Table 4, it is included to directly look at the association between independent and dependent variables of interest that may disappear after adjustment, and a similar approach has been used in previous papers. doi: 10.31662/jmaj.2019-0068. doi: 10.2486/indhealth.ms1139.

Comment:

also, I found the p-value footnotes irrelevant since the Confidence intervals are so much more informative.

Response:

Though we understand that the Confidence intervals are so much more informative, we would like to add a p-value symbol for better visibility because there are a lot of numbers in the table.

Reviewer 5 Report

I appreciate your efforts to improve manuscript. Now, paper is written in a proper manner and it is easy to understand.

I only suggest to include in discussion a last paragraph to reinforce the importance of real world data to reduce vaccine hesitancy. Authors should remark effectiveness of covid vaccine in different waves and safety in case of coadministration (considering WHO recommendations)

·      doi: 10.1016/j.vaccine.2023.07.043

·      doi: 10.3390/ijerph19042282

Author Response

Response to Reviewer 5 comments

  1. Summary

We appreciate your time and effort in reviewing our manuscript and providing us with valuable feedback. Below, you will find our detailed responses to your comments and corresponding revisions, which are highlighted in track changes in the resubmitted manuscript.

  1. Questions for General Evaluation and Reviewer’s Evaluation

Does the introduction provide sufficient background and include all relevant references?

Yes

Are all the cited references relevant to the research?

Yes

Is the research design appropriate?

Yes

Are the methods adequately described?

Yes

Are the results clearly presented?

Yes

Are the conclusions supported by the results?

Can be improved

Response: Thank you again for your evaluation. We revised our manuscript following your comments. We would like you to see the Point-by-point response to Comments and Suggestions for Authors described below.

  1. Point-by-point response to Comments and Suggestions for Authors

Comment:

I appreciate your efforts to improve manuscript. Now, paper is written in a proper manner and it is easy to understand.

Response:

We appreciate you for your valuable feedback.

Comment:

I only suggest to include in discussion a last paragraph to reinforce the importance of real world data to reduce vaccine hesitancy. Authors should remark effectiveness of covid vaccine in different waves and safety in case of coadministration (considering WHO recommendations)

  • doi: 10.1016/j.vaccine.2023.07.043

  • doi: 10.3390/ijerph19042282

Response:

Thank you for your suggestion. As your instruction, we added the following statement in the Discussion section.

“Conducting studies based on real-world data, such as this study, and making the re-sults public is crucial to reduce unnecessary hesitancy towards vaccines. There is also no doubt that vaccination protects many lives against future emerging mutant strains [29].” (Line 303-305)

Also, we added the following description regarding as effectiveness of covid-19 vaccine in different waves and safety in case of coadministration.

“Influenza is currently prevalent in Japan along with COVID-19, and the safety and efficacy of simultaneous vaccination with COVID-19 and influenza vaccines has been recognized by the Government of Japan, as have reports on the safety and efficacy of such vaccination [10], and simultaneous vaccination is now possible in Japan.” (Line80-84)